TECHNICAL RELEASE

# aws-s3-integrity-check: an open-source bash tool to verify the integrity of a dataset stored on Amazon S3

Sonia García-Ruiz[1,2], Regina Hertfelder Reynolds[1,2], Melissa Grant-Peters[1,2], Emil Karl Gustavsson[1,2], Aine Fairbrother-Browne[1,3,4], Zhongbo Chen[1,4], Jonathan William Brenton[1,2] and Mina Ryten[1,2,*]

1 Department of Genetics and Genomic Medicine Research & Teaching, UCL GOS Institute of Child Health, London, UK
2 NIHR Great Ormond Street Hospital Biomedical Research Centre, University College London, London, UK
3 Department of Medical and Molecular Genetics, School of Basic and Medical Biosciences, King's College London, London, UK
4 Department of Neurodegenerative Disease, Queen Square Institute of Neurology, UCL, London, UK

## ABSTRACT

Amazon Simple Storage Service (Amazon S3) is a widely used platform for storing large biomedical datasets. Unintended data alterations can occur during data writing and transmission, altering the original content and generating unexpected results. However, no open-source and easy-to-use tool exists to verify end-to-end data integrity. Here, we present *aws-s3-integrity-check*, a user-friendly, lightweight, and reliable bash tool to verify the integrity of a dataset stored in an Amazon S3 bucket. Using this tool, we only needed ~114 min to verify the integrity of 1,045 records ranging between 5 bytes and 10 gigabytes and occupying ~935 gigabytes of the Amazon S3 cloud. Our *aws-s3-integrity-check* tool also provides file-by-file on-screen and log-file-based information about the status of each integrity check. To our knowledge, this tool is the only open-source one that allows verifying the integrity of a dataset uploaded to the Amazon S3 Storage quickly, reliably, and efficiently. The tool is freely available for download and use at https://github.com/SoniaRuiz/aws-s3-integrity-check and https://hub.docker.com/r/soniaruiz/aws-s3-integrity-check.

**Subjects** Software and Workflows, Software Engineering, Workflows

**Submitted:** 26 April 2023

* Corresponding author. E-mail: mina.ryten@ucl.ac.uk

Preprint submitted at https://doi.org/10.20944/preprints202308.0603.v1

# FINDINGS

## Background

Since the advent of high-throughput next-generation sequencing technologies [1] and with the recent surge of long-read, single-cell, and spatial RNA sequencing [2], biomedical research has become intensely data-driven [3–5]. Indeed, one of the major challenges of the post-genome era has been to store the large data volumes produced by these technologies. Cloud computing providers, such as Amazon Web Services (AWS) [6], play an essential role in addressing this challenge by offering leading security standards, cost-effective data storage, easy data sharing, and real-time access to resources and applications [7–9].

Nevertheless, cloud storage services require a stable network connection to complete a successful data transfer [10]. For instance, network congestion can cause packet loss during

data transmission, producing unintended changes to the data and corrupting the transferred files. To identify faulty data transfers in real-time, Amazon Simple Storage Service (Amazon S3) permits using checksum values through the AWS Command Line Interface (AWS CLI) tool. This approach consists of locally calculating the Content-MD5 or the entity tag (ETag) number associated with the contents of a given file; this checksum value is then inserted within the AWS CLI command used to upload the file to an Amazon S3 bucket. If the checksum number assigned by Amazon S3 is identical to the local checksum calculated by the user, then both local and remote file versions are the same: the file's integrity is proven.

However, this method has disadvantages. First, the choice of the checksum number to calculate (i.e., either the Content-MD5 [11] or the ETag number) depends on the characteristics of the file, such as its size or the server-side encryption selected. This requires the user to evaluate the characteristics of each file independently before deciding which checksum value to calculate. Second, the checksum number needs to be included within the AWS CLI command used to upload the file to an Amazon S3 bucket; thus, the user needs to upload each file individually. Finally, this process needs to be repeated for each file transferred to Amazon S3, which, as the number of files forming a dataset increases, can exponentially increase the time required to complete a data transfer.

To overcome these challenges, we developed *aws-s3-integrity-check*, a bash tool to verify the integrity of a dataset uploaded to Amazon S3. The *aws-s3-integrity-check* tool offers a user-friendly and easy-to-use front-end that requires one single command with a maximum of three parameters to perform the complete integrity verification of all files contained within a given Amazon S3 bucket, regardless of their size and extension. In addition, the *aws-s3-integrity-check* tool provides three unique features: (i) it is used after the data has been uploaded, providing the user with the freedom to transfer the data in batches to Amazon S3 without having to manually calculate individual checksum values for each file; (ii) to complete the integrity verification of all files contained within a given dataset, it only requires the submission of one query to the Amazon S3 application programming interface (API), thus not congesting the network; and (iii) it informs the user of the result from each checksum comparison, providing detailed per-file information. Concerning the latter, *aws-s3-integrity-check* can produce four types of output: (i) if the user does not have read access to the indicated Amazon S3 bucket, the tool produces an error message and stops its execution; (ii) if a given file from the provided folder does not exist within the indicated Amazon S3 bucket, the tool produces a warning message and continues its execution; (iii) if a local file exists within the remote bucket, but its local and remote checksum values do not match, the tool produces a warning message and continues its execution; (iv) if the local file exists within the remote bucket and its local and remote checksum values match, the integrity of the file is marked as proven. All outputs are shown on-screen and stored locally in a log file.

The *aws-s3-integrity-check* tool is freely available for download and use [12, 13], also within a Docker format [14].

## Our approach

Our purpose was to enable the automatic integrity verification of a set of files transferred to Amazon S3, regardless of their size and extension. Therefore, we created the *aws-s3-integrity-check* tool, which: (i) reads the metadata of the totality of files stored within



a given Amazon S3 bucket by querying the Amazon S3 API only once; (ii) calculates the checksum value associated with every file contained within a local folder by using the same algorithm applied by Amazon S3; and (iii) compares local and remote checksum values, informing the user if both numbers are identical and, consequently, if the remote version of the S3 object coincides with its local version.

To identify different file versions, Amazon S3 uses ETag numbers, which remain unalterable unless the file object suffers any change to its contents. Amazon S3 uses different algorithms to calculate an ETag number, depending on the characteristics of the transferred file. More specifically, an ETag number is an MD5 digest of the object data when the file is: (i) uploaded through the AWS Management Console or using the PUT Object, POST Object, or Copy operation; (ii) is plain text; or (iii) is encrypted with Amazon S3-managed keys (SSE-S3). However, if the object has been server-side encrypted with customer-provided keys (SSE-C) or with AWS Key Management Service (AWS KMS) keys (SSE-KMS), the ETag number assigned will not be an MD5 digest. Finally, if the object has been created as part of a `Multipart Upload` or `Party copy` operation, the ETag number assigned will not be an MD5 digest, regardless of the encryption method [15]. When an object is larger than a specific file size, it will be automatically uploaded using multipart uploads. The ETag number assigned to it will combine the different MD5 digest numbers assigned to smaller sections of its data.

In order to match the default values published within the guidelines corresponding to the AWS CLI S3 transfer commands [16], 8 MB is the default multipart chunk size and the maximum file size threshold for *aws-s3-integrity-check* to calculate the ETag number. To automatise the calculation of the ETag value in cases where the file size exceeds the default value of 8 MB, the *aws-s3-integrity-check* tool uses the *s3md5* bash script (version 1.2) [17]. The s3md5 bash script consists of several steps. Using the same algorithm used by Amazon S3, the s3md5 script splits the files larger than 8 MB into smaller parts of that same size and calculates the MD5 digest corresponding to each chunk. Secondly, the s3md5 script concatenates all the bytes from the individual MD5 digest numbers produced, creating a single value and converting it into binary format before calculating its final MD5 digest number. Thirdly, it appends a dash with the total number of parts calculated to the MD5 hash. The resulting number represents the final ETag value assigned to the file. Figure 1 shows a complete overview of the approach followed (please, refer to the Methods section for more details).

## TESTING

### Datasets

To test the *aws-s3-integrity-check* tool, we used 1,045 files stored across four independent Amazon S3 buckets within a private AWS account in the London region (eu-west-2). The rationale behind the inclusion of these four datasets during the testing phase of the *aws-s3-integrity-check* tool was the variability of their project nature, the different file types and sizes they contain, and their availability within two different public repositories (i.e., the European Genome-phenome Archive (EGA) [18] and the data repository GigaDB [19]).

These four datasets occupied ~935 GB of cloud storage space and contained files ranging between 5 Bytes and 10 GB that were individually uploaded to AWS using the AWS CLI `sync` command (version 1) [20]. No specific server-side encryption was indicated while using the `sync` command. In addition, all the configuration values available for the `aws s3 sync`

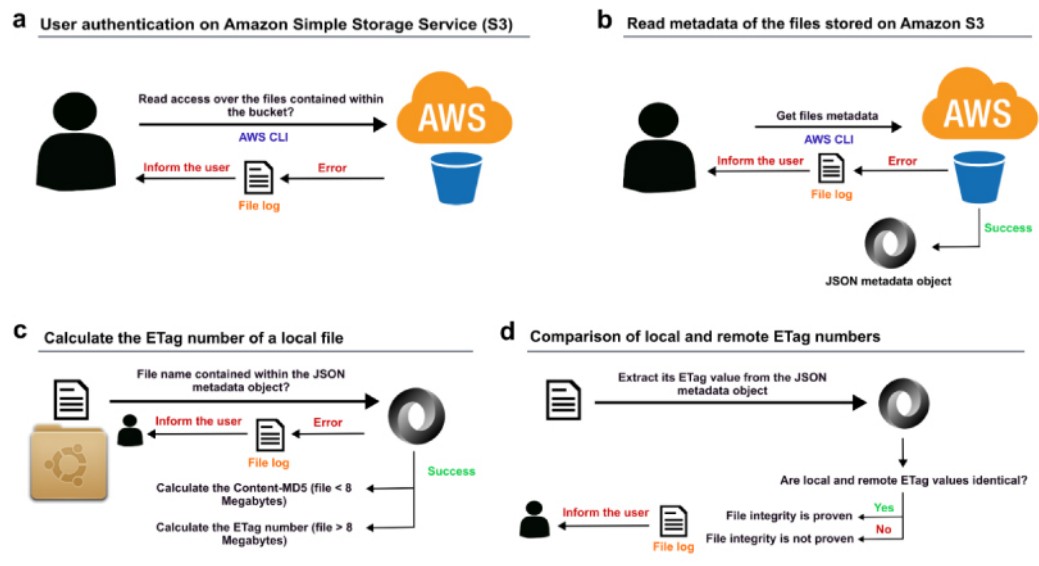

**Figure 1. Overview of the aws-s3-integrity-check tool.**
(A) The *aws-s3-integrity-check* tool verifies if the user has read access to the files within the Amazon S3 bucket indicated by the parameter `[-b|--bucket <S3_bucket_name>]`. (B) The *aws-s3-integrity-check* tool queries the Amazon S3 API to obtain a list with the ETag numbers assigned to all files contained within the S3 bucket indicated. (C) The *aws-s3-integrity-check* tool uses the *s3md5* bash script to calculate the ETag number associated with the contents of each local file contained within the folder indicated by parameter `[-l|--local <path_local_folder>]`. (D) The aws-s3-integrity-check tool compares the local and remote ETag numbers assigned to each local file. The output of each phase of the tool is shown on-screen and logged within a local file.

**Table 1.** Details of the four datasets used during the testing phase of the *aws-s3-integrity-check* tool. All datasets were independently tested. The log files produced by each independent test are available on GitHub [21]. All processing times were measured using the in-built `time` Linux tool (version 1.7) [22]. Processing times refer to the time (in minutes and seconds) required for the *aws-s3-integrity-check* tool to process and evaluate the integrity of the totality of the files within each dataset.

| Amazon S3 bucket | Data origin | Details | Number of files tested | Bucket size | Processing time | Log file |
|---|---|---|---|---|---|---|
| mass-spectrometry-imaging | GigaDB | Imaging-type supporting data for the publication *"Delineating Regions-of-interest for Mass Spectrometry Imaging by Multimodally Corroborated Spatial Segmentation"* [23]. | 36 | 16 GB | real 1m52.193s user 1m8.964s sys 0m24.404s | logs/mass-spectrometry-imaging.S3_integrity_log.2023.07.31-22.59.01.tx |
| rnaseq-pd | EGA | Contents of the EGA dataset EGAS00001006380, containing bulk-tissue RNA-sequencing paired nuclear and cytoplasmic fractions of the anterior prefrontal cortex, cerebellar cortex, and putamen tissues from post-mortem neuropathologically-confirmed control individuals [24]. | 872 | 479 GB | real 62m56.793s user 36m26.604s sys 16m10.548s | logs/rnaseq-pd.S3_integrity_log.2023.07.31-23.02.47.txt |
| tf-prioritizer | GigaDB | Software-type supporting data for the publication *"TF-Prioritizer: a Java pipeline to prioritize condition-specific transcription factors"* [25]. | 6 | 3.7 MB | real 0m15.131s user 0m2.012s sys 0m0.240s | logs/tf-prioritizer.S3_integrity_log.2023.07.31-22.58.33.txt |
| ukbec-unaligned-fastq | EGA | A subset of the EGA dataset EGAS00001003065, containing RNA-sequencing Fastq files generated from 180 putamen and substantia nigra control samples [26]. | 131 | 440 GB | real 51m12.058s user 31m27.348s sys 14m7.084s | logs/ukbec-unaligned-fastq.S3_integrity_log.2023.08.01-01.03.58.txt |

command, which include `max_concurrent_requests`, `max_queue_size`, `multipart_threshold`, `multipart_chunksize`, and `max_bandwidth`, were not changed and used with default values. Details of the four datasets tested are shown in Table 1.



**Table 2.** File types processed during the testing phase of the *aws-s3-integrity-check* tool.

| File type | Description |
|---|---|
| Bam | Compressed binary version of a SAM file that represents aligned sequences up to 128 Mb. |
| Bed | Browser Extensible Data format. This file format is used to store genomic regions as coordinates. |
| Csv | Comma-Separated Values. |
| Docx | File format for Microsoft Word documents. |
| Fa | File containing information about DNA sequences and other related pieces of scientific information. |
| Fastq | Text-based format for storing genome sequencing data and quality scores. |
| Gct | Gene Cluster Text. This is a tab-delimited text format file that contains gene expression data. |
| Gff | General Feature Format is a file format used for describing genes and other features of DNA, RNA, and protein sequences. |
| Gz | A file compressed by the standard GNU zip (gzip). |
| Html | HyperText Markup Language file. |
| Ibd | Pre-processed mass spectrometry imaging (MSI) data. |
| imzML | Imaging Mass Spectrometry Markup Language. Contains raw MSI data. |
| Ipynb | Computational notebooks that can be opened with Jupyter Notebook. |
| Jpg | Compressed image format for containing digital images. |
| JSON | JavaScript Object Notation. Text-based format to represent structured data based on JavaScript object syntax. |
| md5 | Checksum file. |
| Msa | Multiple sequence alignment file. It generally contains the alignment of three or more biological sequences of similar length. |
| Mtx | Sparse matrix format. This contains genes in the rows and cells in the columns. It is produced as output by Cell Ranger. |
| Npy | Standard binary file format in NumPy [27] for saving numpy arrays. |
| Nwk | Newick tree file format to represent graph-theoretical trees with edge lengths using parentheses and commas. |
| Pdf | Portable Document Format. |
| Py | Python file. |
| Pyc | Compiled bytecode file generated by the Python interpreter after a Python script is imported or executed. |
| R | R language script format. |
| Svg | Scalable Vector Graphics. This is a vector file format. |
| Tab | Tab-delimited text or data files. |
| Tif | Tag Image File Format. Tif is a computer file used to store raster graphics and image information. |
| Tsv | Tab-separated values to store text-based tabular data. |
| Txt | Text document file. |
| Vcf | Variant Call Format. Text file for storing gene sequence variations. |
| Xls | Microsoft Excel Binary File format. |
| Zip | A file containing one or more compressed files. |

## File types

Using the *aws-s3-integrity-check* tool, we successfully verified the data integrity of multiple file types detailed in Table 2.

## Testing procedure

We performed two-sided tests. We used the *aws-s3-integrity-check* tool to (i) test the integrity of three datasets uploaded to Amazon S3 and (ii) test the integrity of one dataset downloaded from Amazon S3.

To test the former approach, we downloaded three publicly available datasets corresponding to one EGA project and two GigaDB studies. Firstly, we requested access to the dataset with EGA accession number EGAS00001006380, and, after obtaining access, we downloaded the totality of its files to a local folder. Secondly, we downloaded from the GigaDB File Transfer Protocol server the data files corresponding to two *GigaScience* studies [28, 29] DOI:10.5524/102374 and DOI:10.5524/102379, by using the following Linux commands:

```
$ wget -r
ftp://anonymous@ftp.cngb.org/pub/gigadb/pub/10.5524/102001_103000/102374/*
```

```
$ wget -r
ftp://anonymous@ftp.cngb.org/pub/gigadb/pub/10.5524/102001_103000/102379/*
```

These three datasets (i.e., one EGA dataset and two GigaDB projects) were then uploaded to three different Amazon S3 buckets, which were respectively named *"rnaseq-d"* (EGAS00001006380), *"mass-spectrometry-imaging"* [23], and *"tf-prioritizer"* [25] (Table 1). In all three cases, the data was uploaded to Amazon S3 by using the following `aws s3` command:

```
$ aws s3 sync --profile aws_profile path_local_folder/ s3://bucket_name/
```

To verify that the data contents of the remote S3 objects were identical to the contents of their local version, we then ran the *aws-s3-integrity-check* tool by using the following command structure:

```
$ bash aws_check_integrity.sh [-l|--local <path_local_folder>] [-b|--bucket
<s3_bucket_name>] [-p|--profile <aws_profile>]
```

Next, we used the *aws-s3-integrity-check* tool to test the integrity of a local dataset downloaded from an S3 bucket. In this case, we used data from the EGA project with accession number EGAS00001003065. Once we obtained access to the EGAS00001003065 repository, we downloaded all its files to a local folder. We then uploaded this local dataset to an S3 bucket named *"ukbec-unaligned-fastq"* (Table 1). When the data transfer to Amazon S3 finished, we downloaded these remote files to a local folder by using the S3 command `sync` as follows:

```
$ aws s3 sync --profile aws_profile s3://ukbec-unaligned-fastq/ path_local_folder/.
```

To test that the local version of the downloaded files had identical data contents as their remote S3 version, we ran the *aws-s3-integrity-check* tool employing the following command synopsis:

```
$ bash aws_check_integrity.sh [-l|--local <path_local_folder>] [-b|--bucket
<ukbec-unaligned-fastq>] [-p|--profile <aws_profile>].
```

Finally, we tested whether the *aws-s3-integrity-check* tool could detect any differences between a given local file that had been manually modified and its S3 remote version and inform the user accordingly. Therefore, we edited the file *"readme_102374.txt"* from the dataset [23] and changed its data contents by running the following command:

```
$ (echo THIS FILE HAS BEEN LOCALLY MODIFIED; cat readme_102374.txt) >
readme_102374.tmp && mv readme_102374.t{mp,xt}
```

We then run the *aws-s3-integrity-check* tool employing the following command synopsis:

```
$ bash aws_check_integrity.sh [-l|--local <path_local_folder>] [-b|--bucket
<mass-spectrometry-imaging>] [-p|--profile <aws_profile>].
```

As expected, the *aws-s3-integrity-check* tool was able to detect the differences in data contents between the local and the S3 remote version of the *"readme_102374.txt"* file by producing a different checksum number from the one originally provided by Amazon S3. The error message produced was *"ERROR: local and remote ETag numbers for the file 'readme_102374.txt' do not match.".* The output of this comparison can be checked on the log file *"mass-spectrometry-imaging.S3_integrity_log.2023.07.31-22.59.01.txt"* (Table 1).

The *aws-s3-integrity-check* tool also demonstrated minimal use of computer resources by displaying an average CPU usage of only 2% across all tests performed.



## Testing configuration

The four datasets tested were stored across four Amazon S3 buckets in the AWS London region (eu-west-2) (Table 1). All four S3 buckets had the file versioning enabled and a server-side SSE-S3 encryption key type selected.

The *aws-s3-integrity-check* tool is expected to work for files that have been uploaded to Amazon S3 by following these two uploading criteria:

1. Uploaded by command line using any of the `aws s3` transfer commands, which include the `cp, sync, mv,` and `rm` commands.
2. Using the default values established for the following `aws s3` configuration parameters:

   a. `max_concurrent_requests` - default: 10.
   b. `max_queue_size` - default: 1000.
   c. `multipart_threshold` - default: 8 (MB).
   d. `multipart_chunksize` - default: 8 (MB).
   e. `max_bandwidth` - default: none.
   f. `use_accelerate_endpoint` - default: false.
   g. `use_dualstack_endpoint` - default: false.
   h. `addressing_style` - default: auto.
   i. `payload_signing_enabled` - default: false.

The *aws-s3-integrity-check* tool is expected to work across Linux distributions. With this in mind, testing was performed using an Ubuntu server 16.04 LTS with kernel version 4.4.0-210-generic and an Ubuntu server 22.04.1 LTS (Jammy Jellyfish) with kernel version 5.15.0-56-generic. To remove the operating system barrier, the Dockerized version of the *aws-s3-integrity-check* tool has been made available [14].

## Support

The source code corresponding to the *aws-s3-integrity-check* tool is hosted on GitHub [13]. Also, from this repository, it is possible to create new issues and submit tested pull review requests. Issues have been configured to choose between the "Bug report" and "Feature request" categories, ultimately facilitating the creation and submission of new triaged and labelled entries.

The *aws-s3-integrity-check* tool relies on the *s3md5* bash script (version 1.2) [17] to function. To ensure the availability and maintenance of the s3md5 bash script to users of the *aws-s3-integrity-check* tool, the source s3md5 GitHub repository [17] has been forked and made available [30]. Any potential issues emerging on the s3md5 bash script that may affect the core function of the *aws-s3-integrity-check* tool can be submitted via the Issues tab of the forked s3md5 repository. Any new issue will be triaged, maintained, and fixed on the forked GitHub repository within the "Bug Report" category, before being submitted via a pull request to the project owner.

## Limitations

Here, we presented a novel approach for optimising the integrity verification of a dataset transferred to/from the Amazon S3 cloud storage service. However, there are a few caveats to this strategy. First, the user has to have read/write access to an Amazon S3 bucket.



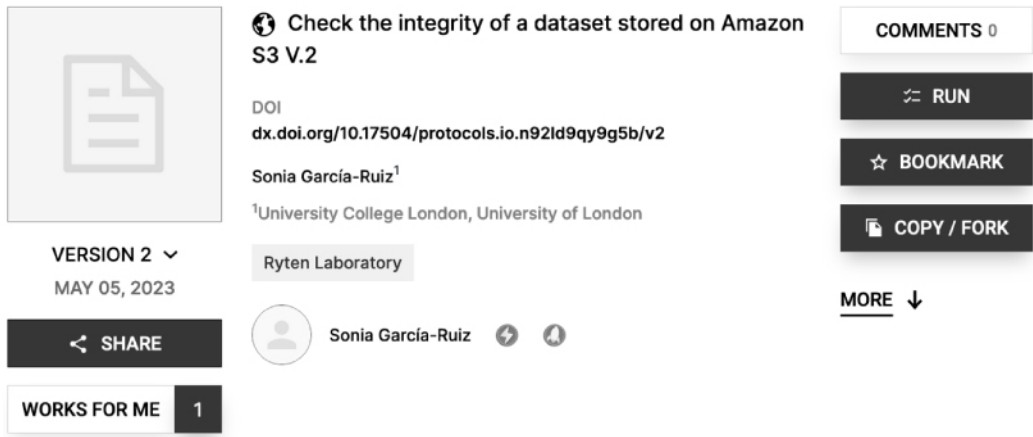

**Figure 2.** A stepwise protocol summarising how to check the integrity of a dataset stored on Amazon S3 [31].
https://www.protocols.io/widgets/doi?uri=dx.doi.org/10.17504/protocols.io.n92ld9qy9g5b/v2

Second, this tool requires that the user selects JavaScript Object Notation (JSON) as the preferred text-output format during the AWS authentication process. Third, the *aws-s3-integrity-check* tool is only expected to work for files that have been uploaded to Amazon S3 using any of the `aws s3` transfer commands (i.e., `cp`, `sync`, `mv`, and `rm`) with all the configuration parameters set to their default values, including `multipart_threshold` and `multipart_chunksize`. In particular, it is essential that the file size thresholds for the file multipart upload and the default multipart chunk size remain at the default 8 MB values. Fourth, the bash version of this tool is only expected to work across Linux distributions. Finally, the Dockerized version of this tool requires three extra arguments to mount three local folders required by the Docker image, which may increase the complexity of using this tool.

## METHODS

A stepwise protocol summarising how to check the integrity of a dataset stored on Amazon S3 is available on protocols.io [31] (Figure 2).

### Main script

The main script is formed by a set of sequential steps whose methods are detailed below.

To parse command options and arguments sent to the *aws-s3-integrity-check* bash tool, we used the Linux built-in function `getops` [32]. The arguments sent corresponded to (i) [`-l`|`--local` `<path_local_folder>`], to indicate the path to the local folder containing the files to be tested; (ii) [`-b`|`--bucket` `<S3_bucket_name>`], to indicate the name of the Amazon S3 bucket containing the remote version of the local files; (iii) [`-p`|`--profile` `<aws_profile>`], to indicate the user's AWS profile in case the authentication on AWS was done using single sign-on (SSO); and (iv) [`-h`|`--help`], to show further information about the usage of the tool.

To test whether the user had read access over the files stored within the Amazon S3 bucket indicated through the argument [`-b`|`--bucket` `<S3_bucket_name>`], we used the AWS CLI command `aws s3 ls` (version 2) [33]. If this query returned an error, the tool informed

```
{ ⊟
    "Contents":[ ⊟
        { ⊟
            "Key":"file_name.extension",
            "LastModified":"2020-04-14T08:49:49+00:00",
            "ETag":"\"etag_number\"",
            "Size":1015292976,
            "StorageClass":"STANDARD",
            "Owner":{ ⊟
                "ID":"ID_number"
            }
        },
        { ⊟
            "Key":"folder_name/file_name.extension",
            "LastModified":"2020-04-24T13:16:29+00:00",
            "ETag":"\"etag_number\"",
            "Size":7205828504,
            "StorageClass":"STANDARD",
            "Owner":{ ⊟
                "ID":"ID_number"
            }
        }
    ]
}
```

**Figure 3.  Structure of the JSON object returned by the AWS CLI function list-objects.**
The information contained within each entry corresponds to the metadata of a given S3 object. The *aws-s3-integrity-check* bash tool used the keys "Key" and "ETag" during the integrity verification of each file.

the user and suggested different AWS authentication options. For the correct performance of this tool, it is required that the user selects JSON as the preferred text-output format during the AWS authentication process.

To obtain the ETag number assigned to the totality of the files contained within the indicated Amazon S3 bucket, we used the AWS CLI command `list-objects` (version 1) [34] as follows:

```
$ aws s3api list-objects --bucket "$bucket_name" --profile "$aws_profile"'
```

In this way, we reduced to one the number of queries performed to the AWS S3 API, known as `s3api`, which considerably reduced the overall network overload. The output of the function `list-objects` was a JSON object (Figure 3).

If the local path indicated through the parameter [-l|--local <path_local_folder>] existed, was a directory, and the user had read access over its contents, the tool looped through its files. For each file within the folder, the *aws-s3-integrity-check* bash tool checked whether the filename was among the entries retrieved within the JSON metadata object and indicated within the *"Key"* field. If that was the case, it meant that the local file existed on the indicated remote Amazon S3 bucket, and we could proceed to calculate its checksum



value. Before calculating the checksum value of the file, the tool evaluated the data content of the file. If it was smaller than 8 MB, the tool calculated its Content-MD5 value by using the function *md5sum* [35, 36] (version 8.25) [37]. However, if the file was larger than 8 MB, it used the function *s3md5* (version 1.2) [17] with the command `"s3md5 8 path_local_file"`.

To obtain the ETag value that Amazon S3 assigned to the tested file the moment it was stored on the remote bucket, we filtered the JSON metadata object using the fields *"ETag"* and *"Key"* and the function `select` (jq library, version jq-1.5-1-a5b5cbe, [38]). We then compared the local and remote checksum values; if the two numbers were identical, the integrity of the local file was proven. We repeated this process for each file in the local folder `[-l|--local <path_local_folder>]`.

To inform the user about the outcome of each step, we use on-screen messages and log this information within a local file in a .txt format. Log files are stored within a local folder named *"log/"* located in the same path in which the main bash script *aws-check-integrity.sh* is located. If a local *"log/"* folder does not exist, the script creates it. Figure 1 shows a complete overview of the approach we followed.

## Docker image

To create the Dockerized version of the *aws-s3-integrity-check* tool (Docker, version 18.09.7, build 2d0083d) [39], we used the Dockerfile shown in Figure 4.

The Dockerized version of the aws-s3-integrity-check tool requires the user to indicate the following additional arguments within the `docker run` command:

- **[-v <path_local_folder>:<path_local_folder>].** This argument is required. This argument requires replacing the strings `[<path_local_folder>:<path_local_folder>]` with the **absolute** path to the local folder containing the local version of the remote S3 files to be tested. This argument is used to mount the local folder as a local volume to the Docker image, allowing Docker to have read access over the local files to be tested. **Important**: the local folder should be referenced by using the absolute path.

  – Example: `-v /data/nucCyt:/data/nucCyt`

- **[-v "$PWD/logs/:/usr/src/logs"].** This argument is required. This argument should not be changed and, therefore, it should be used as it is shown here. It represents the path to the local *logs* folder and is used to mount the local *logs* folder as a local volume to the Docker image. It allows Docker to record the outputs produced during the tool execution.
- **[-v "$HOME/.aws:/root/.aws:ro"]**. This argument is required. This argument should not be changed and, therefore, it should be used as it is shown here. It represents the path to the local folder containing the information about the user authentication on AWS. This parameter is used to mount the local AWS credential directory as a read-only volume to the Docker image, allowing Docker to have read access to the authentication information of the user on AWS.

Next, we present two examples that show how to run the Dockerized version of the *aws-s3-integrity-check* tool. Each example differs in the method used by the user to authenticate on AWS:

```
Code    Blame                                          Raw  ⧉ ⬇ ✏ ▾  <>

  1      FROM ubuntu:18.04

  2

  3      LABEL maintainer="SoniaGR <s.ruiz@ucl.ac.uk>"

  4

  5      RUN apt-get update && apt-get install -y \

  6        --no-install-recommends \

  7        --fix-missing \

  8        ca-certificates \

  9        jq \

 10        xxd \

 11        curl \

 12        unzip

 13

 14      ## Install the AWS CLI dependency (used to interact with AWS services)

 15      RUN curl "https://awscli.amazonaws.com/awscli-exe-linux-x86_64.zip" -o "awscliv2.zip"

 16      RUN unzip awscliv2.zip

 17      RUN ./aws/install

 18

 19      ## Copy and RUN the s3md5 tool dependency (used to calculate ETag number of file multiparts)

 20      RUN mkdir /usr/src/s3md5

 21      COPY ./s3md5/* /usr/src/s3md5/

 22      RUN chmod -R 777 /usr/src/s3md5

 23

 24      ## Copy the aws-s3-integrity-check tool and grant permissions

 25      COPY ./aws_check_integrity.sh /usr/src/aws_check_integrity.sh

 26      RUN chmod 755 /usr/src/aws_check_integrity.sh

 27

 28      ## Create folder to store the logs

 29      RUN mkdir /usr/src/logs

 30      RUN chmod 755 /usr/src/logs

 31

 32      ENTRYPOINT ["/usr/src/aws_check_integrity.sh"]
```

**Figure 4.** **Dockerfile used to Dockerize the *aws-s3-integrity-check* tool.**

Example #1 (if the user has authenticated on Amazon s3 using SSO):

- ```
  docker run -v /data/nucCyt:/data/nucCyt -v "$PWD/logs:/usr/src/logs" -v
  "$HOME/.aws:/root/.aws:ro" soniaruiz/aws-s3-integrity-check:latest -l
  /data/nucCyt/ -b nuccyt -p my_aws_profile
  ```

Example #2 (if the user has authenticated on Amazon s3 using an IAM role (KEY + SECRET)):

- ```
  docker run -v /data/nucCyt:/data/nucCyt -v "$PWD/logs:/usr/src/logs" -v
  "$HOME/.aws:/root/.aws:ro" soniaruiz/aws-s3-integrity-check:latest -l
  /data/nucCyt/ -b nuccyt
  ```

## AVAILABILITY AND REQUIREMENTS
- Project name: **aws-s3-integrity-check: an open-source bash tool to verify the integrity of a dataset stored on Amazon S3**

- Project homepage: https://github.com/SoniaRuiz/aws-s3-integrity-check [12]
- DockerHub URL: https://hub.docker.com/r/soniaruiz/aws-s3-integrity-check
- Protocols.io: https://dx.doi.org/10.17504/protocols.io.n92ld9qy9g5b/v2 [31]
- Operating system: Ubuntu 16.04.7 LTS (Xenial Xerus), Ubuntu 18.04.6 LTS (Bionic Beaver), Ubuntu server 22.04.1 LTS (Jammy Jellyfish).
- Programming language: Bash
- Other requirements:

  – jq (version jq-1.5-1-a5b5cbe, https://stedolan.github.io/jq/)
  – xxd (version 1.10 27oct98 by Juergen Weigert,
     https://manpages.ubuntu.com/manpages/bionic/en/man1/xxd.1.html)
  – s3md5 (https://github.com/antespi/s3md5)
  – AWS Command Line Interface (CLI), (version 2,
     https://docs.aws.amazon.com/cli/latest/userguide/getting-started-install.html)
  – Docker (version 18.09.7, build 2d0083d, https://www.docker.com/)

- License: Apache-2.0 license.

## DATA AVAILABILITY

All datasets used during the testing phase of the *aws-s3-integrity-check* tool are available within the EGA and GigaDB platforms.

The dataset stored within the Amazon S3 bucket *'mass-spectrometry-imaging'* was generated by Guo *et al.* [28] and is available on the GigaDB platform [23].

The dataset stored within the Amazon S3 bucket *'tf-prioritizer'* was generated by Hoffmann *et al.* [29] and is available on the GigaDB platform [25].

The dataset stored within the Amazon S3 bucket *'rnaseq-pd'* was generated by Feleke, Reynolds *et al.* [24] and is available under request from EGA with accession number EGAS00001006380.

The dataset stored within the Amazon S3 bucket *'ukbec-unaligned-fastq'* was a subset of the original dataset generated by Guelfi *et al.* [26], and is available under request from EGA with accession number EGAS00001003065.

The log files produced during the testing phase of the *aws-s3-integrity-check* tool are available at https://github.com/SoniaRuiz/aws-s3-integrity-check/tree/master/logs.

## LIST OF ABBREVIATIONS

Amazon S3: Amazon Simple Storage Service; API: Application Programming Interface; AWS: Amazon Web Services; AWS CLI: AWS Command Line Interface; AWS KMS: AWS Key Management Service; DOI: Digital Object Identifier; EGA: European Genome-phenome Archive; ETag: Entity Tag; JSON: JavaScript Object Notation; MSI: mass spectrometry imaging; SSE-C: Server-side encryption with customer-provided encryption keys; SSE-KMS: Server-side encryption with AWS Key Management Service keys; SSE-S3: Server-side encryption with Amazon S3 managed keys; SSO: Single Sign-On.

## DECLARATIONS

## Competing Interests

The authors declare that they have no competing interests.



## Ethics approval

The authors declare that ethical approval was not required for this type of research.

## Author contributions

SGR implemented the *aws-s3-integrity-check* bash tool, created the manuscript, provided the cloud computing expertise, designed the case study, created the Docker image, and conducted the empirical experiments. RHR provided new ideas for feature development and SSO knowledge. RHR, MG-P, ZC, and AF-B proofread the manuscript. RHR, EKG, JWB, MG-P, AF-B, and ZC helped during the empirical experiments. MR supervised the tool implementation.

## Funding

SGR, RHR, MG-P, JWB and MR were supported through the award of a Tenure Track Clinician Scientist Fellowship to MR (MR/N008324/1). EKG was supported by the Postdoctoral Fellowship Program in Alzheimer's Disease Research from the BrightFocus Foundation (Award Number: A2021009F). ZC was supported by a clinical research fellowship from the Leonard Wolfson Foundation. AF-B was supported through the award of a Biotechnology and Biological Sciences Research Council (BBSRC UK) London Interdisciplinary Doctoral Fellowship.

## Acknowledgements

We acknowledge support from the AWS Cloud Credits for Research (to SGR) for providing cloud computing resources. We acknowledge Antonio Espinosa; James Seward; Alejandro Martinez; Andy Wu; Carlo Mendola; Marc Tamsky for their contributions to the GitHub repository.

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
