## [Editor Report]

Editor’s AssessmentAmazon S3 has become a widely used and reliable platform for storing large datasets, however, when transferring particularly large datasets there is a risk that changes to the original data can occur during transmission. To address this the authors have developed aws-s3-integrity-check, a user-friendly, lightweight and reliable bash tool to verify the integrity of a dataset stored within an Amazon S3 bucket. The testing and review process fixed a few bugs, and limitations on multipart datasets are now clearly presented. So based on this the work provides a useful open-source option to allow verifying the integrity of uploaded datasets.

---

## [Reviewer Report]

Reviewer name and names of any other individual's who aided in reviewerKen ChoDo you understand and agree to our policy of having open and named reviews, and having your review included with the published manuscript. (If no, please inform the editor that you cannot review this manuscript.)YesIs the language of sufficient quality?YesPlease add additional comments on language quality to clarify if neededIs there a clear statement of need explaining what problems the software is designed to solve and who the target audience is? YesAdditional CommentsIs the source code available, and has an appropriate Open Source Initiative license <a href="https://opensource.org/licenses" target="_blank">(https://opensource.org/licenses)</a> been assigned to the code?YesAdditional CommentsAs Open Source Software are there guidelines on how to contribute, report issues or seek support on the code?NoAdditional CommentsIs the code executable?YesAdditional CommentsBut the tool can only be executed in Linux or inside container with base image is Ubuntu, which is contradicted to `Testing was conducted from multiple operating systems.` mentioned in paper.Is installation/deployment sufficiently outlined in the paper and documentation, and does it proceed as outlined?NoAdditional CommentsThe tool's processing time in my case is much longer, dataset 102379 (4.154s), dataset 102374 (6.34s) and dataset EGAS00001003065 (56m20.63s). But the extra time may be due to the location of the bucket region and network speed, so it is not a big concern for the tool's reproducibility. Is the documentation provided clear and user friendly?YesAdditional CommentsIs there enough clear information in the documentation to install, run and test this tool, including information on where to seek help if required?YesAdditional CommentsBut the tool can only be executed in Linux or inside container with base image is Ubuntu, which is contradicted to `Testing was conducted from multiple operating systems.` mentioned in paper.Is there a clearly-stated list of dependencies, and is the core functionality of the software documented to a satisfactory level?YesAdditional CommentsHave any claims of performance been sufficiently tested and compared to other commonly-used packages? NoAdditional CommentsIs test data available, either included with the submission or openly available via cited third party sources (e.g. accession numbers, data DOIs)?YesAdditional CommentsAre there (ideally real world) examples demonstrating use of the software? YesAdditional CommentsBut the tool does not work on multipart upload objects, which makes it unavailable to those multipart uploaded datasets.Is automated testing used or are there manual steps described so that the functionality of the software can be verified?NoAdditional CommentsAny Additional Overall Comments to the Author### What did well 1. The script `aws_check_integrity.sh` can be easily executed using docker. 2. The script `aws_check_integrity.sh` has tested on a range of file size, and the processing time is reasonable. 3. The paper has mentioned its limitation on multipart uploaded datasets, and set the boundary of the tool's use case. 
### Problems 1. The script `aws_check_integrity.sh` can only be executed in Linux OS, which is contradict to `Testing was conducted from multiple operating systems.` mentioned in paper. 
### Suggestions 1. The script `aws_check_integrity.sh` contains 4 errors, 14 warnings after [shellcheck](https://www.shellcheck.net/), which need to be addressed by author. 2. The paper has mentioned its limitation on multipart datasets, but should have also documented in its [repo](https://github.com/SoniaRuiz/aws-s3-integrity-check/) and also in the help message of the script. 3. The script `aws_check_integrity.sh` has typo which need to be addressed by author. 4. To implement tests on the script itself, eg. using the [bats-core](https://github.com/bats-core/bats-core). 5. To remove OS dependency, may be focusing on implementing the tool in Docker and document its details usage in paper and also in [repo](https://github.com/SoniaRuiz/aws-s3-integrity-check/). 
RecommendationMinor Revisions

---

## [Reviewer Report]

Reviewer name and names of any other individual's who aided in reviewerJens JensenDo you understand and agree to our policy of having open and named reviews, and having your review included with the published manuscript. (If no, please inform the editor that you cannot review this manuscript.)YesIs the language of sufficient quality?YesPlease add additional comments on language quality to clarify if neededA few oddities here and there but nothing that hinders understandingIs there a clear statement of need explaining what problems the software is designed to solve and who the target audience is? NoAdditional CommentsSee comments to authorsIs the source code available, and has an appropriate Open Source Initiative license <a href="https://opensource.org/licenses" target="_blank">(https://opensource.org/licenses)</a> been assigned to the code?YesAdditional CommentsCode is shell script on githubAs Open Source Software are there guidelines on how to contribute, report issues or seek support on the code?YesAdditional CommentsStandard github practice (pull requests)Is the code executable?YesAdditional CommentsShell scriptsIs installation/deployment sufficiently outlined in the paper and documentation, and does it proceed as outlined?YesAdditional CommentsCode has support for detecting when the AWS environment is not correctly set upIs the documentation provided clear and user friendly?YesAdditional CommentsThe paper is very detailed... which I quite like, as often in computing papers people gloss over important detailsIs there enough clear information in the documentation to install, run and test this tool, including information on where to seek help if required?YesAdditional CommentsIs there a clearly-stated list of dependencies, and is the core functionality of the software documented to a satisfactory level?YesAdditional CommentsHave any claims of performance been sufficiently tested and compared to other commonly-used packages? YesAdditional CommentsPerformance should be mostly dependent on the local checksumming toolIs test data available, either included with the submission or openly available via cited third party sources (e.g. accession numbers, data DOIs)?NoAdditional CommentsNot applicableAre there (ideally real world) examples demonstrating use of the software? YesAdditional CommentsIs automated testing used or are there manual steps described so that the functionality of the software can be verified?NoAdditional CommentsNot applicableAny Additional Overall Comments to the AuthorThis paper describes a tool for verifying the integrity of data uploaded to Amazon S3, with specific applications to NGS data. The paper is very, very detailed, which in many ways are quite welcome, as it would make it suitable for beginners in the field as well, and authors often gloss over important details. The downside is that the paper is quite long for what it presents, and with some repetition.  As is standard practice, integrity checking is done at the binary level, with checksums. This means that different encodings of the same file will mismatch (as a simple example, a CSV file may have LF or CRLF line endings), or the file could be compressed in storage, as it would be on tape, but you want to compare the checksum of the uncompressed file. Using checksum alone is certainly sufficient for the simple application described here where the purpose is only to verify that an upload has completed.  If a checksum is transmitted with the file, only a single file can be uploaded. The other problem with this approach is that the file has to be read twice: once to calculate its checksum, and once to upload the file.  Despite the much-publicised cryptographic weaknesses of MD5, it is certainly sufficient to do non-cryptographic integrity checking -- meaning we protect against accidental modification of the file, not modification by a malicious adversary.  Coming from disciplines such as high energy physics, where data integrity is the primary security requirement, we checksum obsessively. The most common failure mode we see is truncated files: the transfer, for some reason, was interrupted and not resumed or retried (which is normally done automatically). For a single stream transfer we would calculate the checksum as the data comes in over the transfer interface -- because it is cheap to do (ie doesn't involve extra disk access) -- but would still need to eventually do a "deep" checksum (of the file on disk) to check that the file was not just transferred but also written correctly.  Splitting the file and into chunks could be a bit inefficient, though it is convenient for a shell script. A third party tool ("s3md5") takes a chunk from the file (using dd and piping it to md5sum) one chunk at a time. This is fine as no temporary copy is created, and reads the file sequentially, though it requires an extra read of the file (one for checksums, one for the transfer) which is not too expensive if the local file is cached in RAM, as it would likely be if it not too large.  The English is generally very good, though with a few odditites here and there (such as "Per each" => "For each")  Specific comments:  Seems odd to have an abstract in three sections 
"private AWS account in London ..." is repeated. 
"reverse directionality of the data transfer" ?? What does that mean - download from the bucket?  While I appreciate the beginner friendly details, the screenshot of the modified file (Fig 2) does not really add much value to the paper - you could even do the modification with shell commands:  (echo THIS FILE HAS BEEN LOCALLY MODIFIED; cat readme_102374.txt) >readme_102374.tmp && \ mv readme_102374.t{mp,xt}  You may get better performance indication by using the `time` tool  For shell scripts, I'd say testing across different versions of the same OS (Ubuntu 16.04 and 22.04) is not essential - it would be better to have (say) Ubuntu and CentOS though again shell scripts are very portable in POSIX environments  So the file object does not store the chunk size that was used to calculate the ETag? That seems like a strange omission.  Being a bit technical, in your Dockerfile, you don't want to run too many independent apt-gets: each time you add a layer to the image. It would be more efficient to consolidate some, eg  RUN apt-get install x RUN apt-get install y RUN apt-get install z  becomes  RUN apt-get install x y z
RecommendationAccept

---

## [Reviewer Report]

Reviewer name and names of any other individual's who aided in reviewerTomasz NeugebauerDo you understand and agree to our policy of having open and named reviews, and having your review included with the published manuscript. (If no, please inform the editor that you cannot review this manuscript.)YesIs the language of sufficient quality?YesPlease add additional comments on language quality to clarify if neededIs there a clear statement of need explaining what problems the software is designed to solve and who the target audience is? YesAdditional CommentsIs the source code available, and has an appropriate Open Source Initiative license <a href="https://opensource.org/licenses" target="_blank">(https://opensource.org/licenses)</a> been assigned to the code?YesAdditional CommentsAs Open Source Software are there guidelines on how to contribute, report issues or seek support on the code?NoAdditional CommentsThis is hosted on GitHub, and there is an "Issues" forum on the repository, but it has no posted open or closed issues. I would encourage the authors to add a statement about support and how to submit issues .Is the code executable?Unable to testAdditional CommentsI did not execute the code, because I do not have an AWS account.Is installation/deployment sufficiently outlined in the paper and documentation, and does it proceed as outlined?Unable to testAdditional CommentsIs the documentation provided clear and user friendly?YesAdditional CommentsIs there enough clear information in the documentation to install, run and test this tool, including information on where to seek help if required?YesAdditional CommentsIf a user encounters errors with s3md5, should they contact the author of s3md5 or aws-s3-integrity-check? This tool relies on s3md5 for a significant portion of the functionality, but s3md5 doesn't look like it has been updated or maintained.Is there a clearly-stated list of dependencies, and is the core functionality of the software documented to a satisfactory level?YesAdditional CommentsHave any claims of performance been sufficiently tested and compared to other commonly-used packages? YesAdditional CommentsIs test data available, either included with the submission or openly available via cited third party sources (e.g. accession numbers, data DOIs)?YesAdditional CommentsAre there (ideally real world) examples demonstrating use of the software? YesAdditional CommentsIs automated testing used or are there manual steps described so that the functionality of the software can be verified?YesAdditional CommentsAny Additional Overall Comments to the AuthorOverall, I think this is a useful tool that offers important functionality for digital preservation of datasets. The manuscript describes the functionality of the tool clearly, and offers detailed instructions and test results. I recommend accepting this article, but suggest a minor revision to address the following :  One weakness, alluded to in the limitations section with the following comment: "Fourth, this tool has not been tested using server-side encryption different from the default option using an SSE-S3 key." I would suggest a clearer statement here instead, as it seems that the tool will not work for server-side encryption different from SSE-S3. It doesn't seem like it's just a matter of not testing. In summary, it would be important to include a statement about whether or not the authors would expect that the tool would work for the objects described in the AWS documentation as follows: 
"If an object is created by the PUT Object, POST Object, or Copy operation, or through the AWS Management Console, and that object is encrypted by server-side encryption with customer-provided keys (SSE-C) or server-side encryption with AWS Key Management Service (AWS KMS) keys (SSE-KMS), that object has an ETag that is not an MD5 digest of its object data." (https://docs.aws.amazon.com/AmazonS3/latest/userguide/checking-object-integrity.html)  The other significant issue is with the sustainability and support of this solution, due to the reliance on s3md5, a separate script authored by someone other than the authors, developed about 10 years ago and last updated in 2016. There is an open issue with s3md5 from March 2019 (https://github.com/antespi/s3md5/issues/11) that has not received any response or comment. Is s3md5 supported or maintained? Will the authors of aws-s3-integrity-check commit to supporting questions about it, and potentially a fork of s3md5 if necessary, since they rely on it for such a significant part of the integrity check?  
RecommendationMinor Revisions